# Corrosion and Erosion Wear Behaviors of HVOF-Sprayed Fe-Based Amorphous Coatings on Dissolvable Mg-RE Alloy Substrates

**DOI:** 10.3390/ma16145170

**Published:** 2023-07-22

**Authors:** Jun Yang, Yijiao Sun, Minwen Su, Xueming Yin, Hongxiang Li, Jishan Zhang

**Affiliations:** 1National Engineering Laboratory for Exploration and Development of Low-Permeability Oil & Gas Fields, Xi’an 710018, China; yangjun4508@163.com (J.Y.); sumw@cnpc.com.cn (M.S.); 2Changqing Downhole Technology Company, Chuanqing Drilling Engineering Co., Ltd., Xi’an 710018, China; cqjx_yinxm@cnpc.com.cn; 3State Key Laboratory for Advanced Metals and Materials, University of Science and Technology Beijing, Beijing 100083, China; sunyijiao0920@126.com

**Keywords:** dissolvable magnesium alloys, Fe-based amorphous coatings, high-velocity oxygen–fuel (HVOF) spraying, corrosion resistance, erosion wear resistance

## Abstract

To suppress the corrosion and erosion wear of dissolvable magnesium alloy ball seats in wellbores, Fe-based amorphous coatings were deposited on dissolvable Mg-RE alloy substrates, and their microstructure, corrosion behavior, and erosion wear behavior were studied. The thickness of Fe-based amorphous coatings on dissolvable Mg-RE alloy substrates can reach 1000 μm without any cracks, and their porosity and amorphous contents are 0.79% and 86.8%, respectively. Although chloride ions will damage the compactness and protective efficacy of passive films, Fe-based amorphous coatings still maintain low corrosion current density (3.31 μA/cm^2^) and high pitting potential (1 V_SCE_) in 20 wt% KCl solution. Due to their higher hardness, the erosion wear resistance of Fe-based amorphous coatings is about 4.16 times higher than that of dissolvable Mg-RE alloy substrates when the impact angle is 30°. Moreover, the erosion rates of Fe-based amorphous coatings exhibit a nonlinear relationship with the impact angle, and the erosion rate reaches the highest value when the impact angle is 60°. The erosion wear mechanisms of Fe-based AMCs vary with the impact angles, including cutting, delamination, splat fracture, and deformation wear. This work can provide effective guidance for the corrosion and wear protection of plugging tools made from dissolvable magnesium alloy.

## 1. Introduction

Oil and gas, as vital national strategic resources, constitute indispensable energy sources for industrial production and serve as the primary feedstocks for various chemical products [1]. With economic growth and the depletion of conventional hydrocarbon resources, unconventional hydrocarbon resources with enormous reserves have garnered widespread attention [2,3]. Multistage ball-drop-activated fracturing systems can substantially augment reservoir permeability and reduce fluid viscosity, allowing economically viable extraction of unconventional hydrocarbon resources. Ball seats—one of the crucial components of fracturing sliding sleeves—are used to open the sandblasting ports for fracturing in conjunction with the fracturing balls [4,5]. Conventional ball seats, typically made of steels, need to be drilled out to avoid hindrance [6]. The novel ball seats crafted with dissolvable magnesium alloys can be degraded after fracturing operations, leaving unobstructed wellbores without drilling or salvage [1,7]. Despite their numerous benefits, issues such as the poor corrosion resistance and low hardness of dissolvable magnesium alloys have made the dissolvable ball seats vulnerable to corrosion from groundwater with variable salinity and to erosion wear from sand-bearing fracturing fluids, thereby causing destruction of the ball seats and the malfunction of the fracturing sliding sleeves [8,9,10]. Therefore, it is essential to prepare erosion- and corrosion-resistant coatings on the dissolvable magnesium alloy ball seats to guarantee their intact condition before fracturing operations.

Numerous methods have been developed for the preparation of protective coatings on magnesium alloys, including chemical conversion [11], plasma electrolytic oxidation (PEO) [12], anodizing [13], electrochemical plating [14], polymer coatings [15], and vapor deposition [16]. Nonetheless, these coatings are inadequate to withstand the corrosion and erosion wear in extreme subsurface environments, on account of their low hardness and low thickness. For instance, PEO coatings have been prepared to delay the degradation of dissolvable magnesium matrix composites, but their thickness (<20 μm) is insufficient to protect the substrate against erosion wear [17]. In contrast, hard alloy coatings, such as WC-Co coatings and WC-Cr_3_C_2_-Ni coatings, can be prepared via thermal spraying technology, and these coatings can significantly improve the erosion wear resistance of the fracturing tools due to their high hardness and considerable thickness [18,19]. However, the multiphase structures and microdefects present in hard alloy coatings pose challenges in providing optimal protection for magnesium alloy substrates, as they typically demonstrate poor corrosion resistance.

Due to their amorphous structure, abundance of passivity promoters/dissolution moderators, and inexpensive main element, thermal-sprayed Fe-based amorphous metallic coatings (AMCs) have the advantages of high corrosion resistance, exceptional wear resistance, and cost-effectiveness [20]. Zhang et al. reported that the corrosion rate of an Fe-based AMC was only one-fourth that of 316 stainless steel in simulated deep-sea environments [21]. Similarly, in sulfate-reducing bacteria (SRB)-inoculated Postgate’s C medium, an Fe-based AMC exhibited lower corrosion current density than 304 stainless steel and X80 pipeline steel, because of its superior pitting resistance and antibacterial properties [22]. The remarkable corrosion resistance of Fe-based AMCs can be attributed to the formation of dense passive films on their surfaces. Moreover, these passive films serve as a barrier between the corrosion solution and the coatings, significantly hindering the progression of corrosion [23]. Owing to their high strength, high hardness, and excellent corrosion resistance, Fe-based AMCs hold remarkable potential for widespread application as erosion-resistant materials [24,25,26,27]. It has been reported that although the corrosion resistance of 304 stainless steel is higher than that of Fe-based AMC, the Fe-based AMC exhibits higher erosion resistance [24]. The current density responses of an Fe-based AMC and 304 stainless steel were measured and analyzed during slurry erosion, and the higher erosion resistance of Fe-based AMCs was attributed to their higher hardness and better re-passivation ability [25,26]. Moreover, similar to 304 stainless steel, the erosion rates of Fe-based AMCs increased with the increase in flow velocity, NaCl concentration, and sand particle size. The erosion rates of 304 stainless steel were the highest at an impact angle of 45°, while the erosion rates of Fe-based AMCs continued to increase with the increase in the impact angle, with the highest erosion rates occurring at an impact angle of 90° [27]. If Fe-based AMCs can be deposited on dissolvable magnesium alloy ball seats, they would be expected to protect the dissolvable ball seats from corrosion and erosion prior to fracturing operations, ensuring the reliable operation of the fracturing sliding sleeves. However, the service status of Fe-based AMCs on dissolvable magnesium alloys in downhole environments remains unknown, especially the corrosion behavior in groundwater with variable salinity and the erosion wear behavior in sand-bearing fracturing fluids.

To this end, Fe-based AMCs were prepared on dissolvable magnesium–rare-earth (Mg-RE) alloy substrates via high-velocity oxygen–fuel (HVOF) spraying technology. The corrosion resistance of Fe-based AMCs in groundwater with variable salinity and the erosion wear resistance of Fe-based AMCs at different impact angles were investigated systematically in this study. In addition, the corrosion and erosion morphologies of Fe-based AMCs, as well as the physical and chemical properties of passive films on Fe-based AMCs, were also characterized to study the corrosion and erosion wear mechanisms on Fe-based AMCs.

## 2. Materials and Methods

Fe-based AMCs with Ni60 interlayers on dissolvable Mg-RE alloy substrates were prepared using an HVOF spraying device (Lijia Thermal Spraying Machinery Co., Zhengzhou, China). Fe-based amorphous powders (Fe_48.8_Cr_23.4_Mo_19.8_Si_5_C_2.1_B_0.9_, wt%) with an average size of 20 μm and Ni60 powders (Ni_62.6_Cr_19_Mo_3.2_Fe_4.5_Cu_2.3_Si_3.7_C_0.7_B_4.0_, wt%) with an average size of 36 μm were selected as feedstock powders, and Mg-Gd-Y-Zn-Cu alloy plates of 140 mm × 150 mm × 10 mm in size were employed as the substrates. Before spraying, the surface of the plates was meticulously cleaned to remove any greasy dirt and sandblasted to improve their surface roughness. The spraying parameters for the Fe-based AMCs and Ni60 interlayers are summarized in Table 1.

Phase analyses of Fe-based amorphous powders and Fe-based AMCs were conducted using a diffractometer (XRD, Bruker Corporation, Billerica, MA, USA) equipped with a Cu target material. The data were collected in the range of 2Theta values between 10° and 90°, at a scan speed of 0.0667°/s. The thermophysical properties of the powders and coatings were determined via differential scanning calorimetry (DSC, TA Instruments, New Castle, DE, USA), and the DSC data were analyzed using TA Universal Analysis 2000 software. The amorphous contents of the Fe-based AMCs were calculated by dividing the crystallization exothermic enthalpy of Fe-based amorphous powders by that of Fe-based AMCs. The microstructure and composition analysis of the Fe-based AMCs, Ni60 interlayers, and dissolvable Mg-Gd alloy substrate were examined using scanning electron microscopy (SEM, Carl Zeiss AG, Oberkochen, Germany) and energy-dispersive spectrometry (EDS). The microscopic morphology and phase structures of the Fe-based AMCs and Ni60 interlayers were also analyzed via high-resolution transmission electron microscopy (TEM, FEI Company, Hillsboro, OR, USA). The chemical composition of the passive films that formed on the Fe-based AMCs was detected and analyzed by X-ray photoelectron spectrometry (XPS, Kratos Analytical, Manchester, UK) with an Al target material.

The porosity of the Fe-based AMCs and Ni60 interlayers was calculated using Image Pro 6.0 software, and more than 10 cross-sectional SEM images were analyzed with a magnification of 500 times to obtain accurate and reliable data. To determine the microhardness of the Fe-based AMCs, Ni60 interlayers, and dissolvable magnesium alloy substrates, a Vickers hardness meter (Yashitejiu Instruments Corporation, Hangzhou, China) was used, with an applied load of 100 g (0.98 N). Each sample was tested at least 15 times, and their average value was taken as the average microhardness of the samples.

The electrochemical properties of the Fe-based AMCs were assessed using an electrochemical workstation (Corrtest Instruments Corporation, Wuhan, China). The samples were connected to a copper guide and sealed with epoxy resin, exposing only 1 cm^2^ of polished area. Aqueous KCl solutions with different concentrations were utilized to simulate groundwater with variable salinity. Potentiodynamic polarization tests, potentiostatic polarization tests, electrochemical impedance spectroscopy (EIS) tests, and Mott–Schottky measurements were conducted after the open-circuit potential was stabilized. The scanning rate of the potentiodynamic polarization tests was 0.5 mV/s, and the polarization potential of the potentiostatic polarization tests was 0.3 V_SCE_. In the EIS tests, the sinusoidal voltage signal was 10 mV, the frequency ranged from 10^−2^ Hz to 10^5^ Hz, and the EIS data were analyzed and processed using ZView 2.8 software. Mott–Schottky measurements of the Fe-based AMCs were carried out with a signal frequency of 1000 Hz and a scanning rate of 25 mV/s. The erosion wear resistance of the Fe-based AMCs and dissolvable magnesium alloy substrates was evaluated using an erosion tester (ZKGX Research Institute of Chemical Technology, Beijing, China). The testing samples of 20 mm × 20 mm × 10 mm in size were cleaned and polished. In order to investigate the effect of the impact angle on erosion wear, the angle was set at 10°, 20°, 30°, 45°, 60°, 75°, and 90°. Clean water was utilized as the solution, while quartz sands with a particle size of 150 μm were selected as the abrasive material. The flow velocity of the slurry was set to 40 m/s, and the sand concentration was maintained at 10 wt%. The erosion wear morphology of the Fe-based AMCs was observed using an ultra-depth-of-field microscope (KEYENCE, Osaka, Japan), and the volume loss was statistically calculated.

## 3. Results and Discussion

### 3.1. Microstructure of the Fe-Based AMC

The microstructures of the Fe-based AMC, Ni60 interlayer, and dissolvable Mg-RE alloy substrate are shown in Figure 1. The coating on the dissolvable Mg-RE alloy substrate was composed of two parts, i.e., the inner layer was a Ni60 interlayer with a thickness of 500 μm, and the outer one was an Fe-based AMC with a thickness of 1000 μm. Composed of stacked splats, both the Fe-based AMC and Ni60 interlayer had typical layered structures. Moreover, the porosity of the Fe-based AMC and Ni60 interlayer was measured to be 0.79% and 0.98%, respectively. Based on the SEM images of the Fe-based AMC and Ni60 interlayer, the inter-splat regions in the Fe-based AMC are more pronounced than those in the Ni60 interlayer, indicating that the Ni60 alloy powders—as self-fluxing alloys—possess excellent wettability and oxidation resistance [28]. As shown in Figure 1c, there is only one diffuse halo in the selected-area electron diffraction (SAED) pattern of the inter-splat region, confirming the amorphous structure of both the splats and the oxides in the inter-splat regions. Correspondingly, there are a diffuse halo and some crystal diffraction patterns in the SAED pattern of the Ni60 splats, so the Ni60 interlayer is composed of both amorphous and crystalline phases, as shown in Figure 1e. In line with this result, there are many crystalline phases varying in size from 20 nm to 200 nm in the amorphous matrix, according to the TEM image of the Ni60 splats. As depicted in Figure 1f,g, the dissolvable Mg-RE alloy substrate comprises a block long-period stacking-ordered (LPSO) phase, plate LPSO phase, RE-rich phase, and α-Mg phase. Due to the high potential difference between the LPSO phase/RE-rich phase and the α-Mg phase, galvanic corrosion will take place on the dissolvable Mg-RE alloy in the corrosion solution, resulting in a high degradation rate [29,30].

Figure 2a,b present the XRD patterns and DSC curves of the Fe-based AMC and Fe-based amorphous powders, respectively. According to the XRD pattern, there is only a diffuse peak in the 2θ range of 34–56°, indicating that both the Fe-based AMC and the Fe-based amorphous powders exhibit an amorphous structure. Based on the DSC curves, there are four crystallization exothermic peaks in the range of 600–875 °C. Fitting analysis reveals that the crystallization exothermic enthalpies of the Fe-based AMC and Fe-based amorphous powders are about 151 J/g and 174 J/g, respectively, and the amorphous contents of the Fe-based AMC (relative to the powders) can be calculated to be 86.8%. The microhardness distribution from the dissolvable Mg-RE alloy substrate to the Fe-based AMC is illustrated in Figure 2c. The average microhardness of the dissolvable Mg-RE alloy substrate, Ni60 interlayer, and Fe-based AMC is about 118 HV_0.1_, 620 HV_0.1_, and 803 HV_0.1_, respectively. Due to their high hardness, Fe-based AMCs are expected to become an ideal material for protection against erosion. The presence of the Ni60 interlayer can effectively alleviate the performance difference between Fe-based AMCs and dissolvable Mg-RE alloy substrates.

### 3.2. Corrosion Behavior of the Fe-Based AMC

Figure 3a displays the potentiodynamic polarization curves of Fe-based AMCs on dissolvable Mg-RE alloy substrates in KCl solutions with diverse concentrations. The corresponding corrosion data garnered from these potentiodynamic polarization curves are presented in Table 2. The corrosion potentials (E_corr_) of the Fe-based AMCs lie within the range of −0.3 V_SCE_ to −0.25 V_SCE_, regardless of the KCl concentration, demonstrating that the KCl concentration remains inconsequential in terms of the corrosion potentials. Concurrently, the corrosion current densities of the Fe-based AMCs exhibit an upward trend as the KCl concentration increases, suggesting a reduction in the corrosion resistance of the Fe-based AMCs. Furthermore, the pitting potentials of the Fe-based AMCs consistently remain around 1 V_SCE_ when the KCl concentration rises from 0.05 wt% to 20 wt%, denoting exceptional pitting resistance in high-concentration KCl solutions. According to the potentiodynamic polarization curves, the passivation process for Fe-based AMCs consists of two distinct stages: The initial stage, known as the stable passivation zone, is marked by low polarization potentials. Within this zone, the passive films formed on the Fe-based AMCs are compact and stable, and the passivation current densities remain almost constant. Conversely, the subsequent stage, termed the over-passivation zone, is characterized by high polarization potentials. In this zone, the compactness of the passive films decreases due to the elevated polarization potential, leading to the initiation of metastable pitting corrosion and severe fluctuations in current densities [31]. The critical potential, identified as the transition between these two stages, was ascertained to be about 0.683 V_SCE_ in 0.05 wt% KCl solution, and it decreased with the increase in the KCl concentration. Notably, the critical potential when the KCl concentration was 20 wt% was estimated to be about 0.099 V_SCE_.

The Nyquist plots, Bode impedance plots, and Bode phase-angle plots of the Fe-based AMCs in KCl solutions with various concentration are shown in Figure 3b–d, respectively. The Nyquist plots of the Fe-based AMCs exhibit two capacitive rings, with the size of the capacitive rings decreasing as the KCl concentration increases. The equivalent circuit of the Fe-based AMCs, as depicted in Figure 3c, comprises solution resistance (R_s_), the sum of the coating resistance and solution resistance in pores (R_c_), capacitance of Fe-based AMCs (CPE_c_), charge-transfer resistance at the Fe-based AMCs–corrosion solution interface (R_t_), and double-layer capacitance of the Fe-based AMCs–corrosion solution interface (CPE_dl_) [32]. The fitting parameters of the EIS data recorded in KCl solution with various concentrations are listed in Table 3. In general, the corrosion resistance of Fe-based AMCs can be evaluated by comparing the values of R_c_ and R_t_, and the higher the resistance, the higher the corrosion resistance of the Fe-based AMCs. In 0.05 wt% KCl solution, the R_c_ and R_t_ of the Fe-based AMC were 337,440 Ω·cm^2^ and 999,910 Ω·cm^2^, respectively. However, as the KCl concentration increased to 20 wt.%, the R_c_ and R_t_ decreased to 38,112 Ω·cm^2^ and 31,655 Ω·cm^2^, respectively. This decrease in R_c_ and R_t_ illustrates the significant deterioration in the corrosion resistance of Fe-based AMCs under increasing KCl concentrations. Furthermore, the capacitance values of CPE_c_ and CPE_dl_ increase with the increase in KCl concentration, because of the damage caused by Cl^−^ ions to the passive films and the increase in the true corrosion areas caused by solution infiltration and corrosion [33].

In a corrosion solution, passive films with semiconductor properties will be formed on Fe-based AMCs to prohibit the direct interaction between the coating and the corrosion solution, slow down the charge transfer, and improve the corrosion resistance [34]. The protective efficacy of passive films is contingent upon their carrier density. Specifically, passive films with low carrier density possess superior compactness and enhanced protective capabilities [35]. In KCl solutions with various concentrations, the Mott–Schottky curves of the passive films formed on Fe-based AMCs are illustrated in Figure 4a. Notably, the Mott–Schottky curves highlight two distinct linear regions: one with a positive slope from −0.4 V_SCE_ to 0.2 V_SCE_, and another one with a negative slope from 0.5 V_SCE_ to 0.8 V_SCE_. Therefore, bipolar passive films can be formed on Fe-based AMCs, regardless of KCl concentration. That is, the passive films exhibit the combined characteristics of both p-type and n-type semiconductors. The donor (*N_D_*) or acceptor (*N_A_*) concentration of the passive films can be calculated from the space-charge capacitance *C_SC_*, and the formula is as follows [36]:$$C_{SC}^{-2}=\frac{2}{eN_{D}\varepsilon \varepsilon_{0}}\left(E-E_{FB}-\frac{kT}{e}\right)\;\;\;\;\;\;\left(n-type\right)$$
$$C_{SC}^{-2}=\frac{-2}{eN_{A}\varepsilon \varepsilon_{0}}\left(E-E_{FB}-\frac{kT}{e}\right)\;\;\;\;\;\;\left(p-type\right)$$
where *C_SC_* represents the space-charge capacitance, *e* stands for the electronic charge, *ε*_0_ denotes the vacuum permittivity, *ε* represents the relative permittivity of the passive film (15.6), *k* denotes the Boltzmann constant, *T* represents the temperature (Kelvin), *E* stands for the film formation potential, and *E_FB_* refers to the flat-band potential. Figure 4b shows the donor concentration and acceptor concentration of the passive films formed on Fe-based AMCs in KCl solutions with various concentrations. In 0.05 wt% KCl solution, the donor concentration and acceptor concentration of passive films formed on Fe-based AMCs were 2.13 × 10^21^ cm^−3^ and 5.86 × 10^21^ cm^−3^, respectively. As the KCl concentration increased to 20 wt%, the donor concentration and acceptor concentration of the passive films were 1.50 × 10^23^ cm^−3^ and 7.12 × 10^22^ cm^−3^, respectively. Therefore, the carrier concentration of the passive films increases with the increase in KCl concentration, indicating a reduction in the compactness and protective capabilities of the passive films.

To further investigate the influence of Cl^−^ ions on the passive films, the surface chemical compositions of Fe-based AMCs immersed in 3 wt% or 20 wt% KCl solution for 30 d were examined. The XPS spectra of the passive films that formed on the Fe-based AMCs are depicted in Figure 5. Whether in 3 wt% KCl solution or 20 wt% KCl solution, the XPS spectra of Fe 2p contain four groups of bimodal peaks, namely, Fe^0^, FeO, Fe_2_O_3_, and FeOOH; the XPS spectra of Cr 2p consist of three groups of bimodal peaks, including Cr^0^, Cr_2_O_3_, and Cr(OH)_3_. Additionally, the XPS spectra of Mo 3d comprise three groups of bimodal peaks, which are Mo^0^, MoO_2_, and MoO_3_. Consequently, with the increase in the KCl concentration, the constituent phases of the passive films formed on the Fe-based AMCs remain unaltered, but the relative contents of the constituent phases are changed significantly. Based on the XPS analysis, the Fe_ox_/Fe_hy_ ratio of the passive film formed in 3 wt% KCl solution was 2.87, while it decreased to 2.76 when the KCl concentration increased to 20 wt%. As the KCl concentration increased from 3 wt% to 20 wt%, the Cr_ox_/Cr_hy_ ratio in the passive films decreased from 2.59 to 1.25, while the Mo^4+^/Mo^6+^ ratio increased from 1.27 to 1.93.

The chemical compositions of the passive films formed in 3 wt% and 20 wt% KCl solutions are shown in Figure 6. The passive films on the Fe-based AMCs, irrespective of KCl concentration, were composed of FeO, Fe_2_O_3_, FeOOH, Cr_2_O_3_, Cr(OH)_3_, MoO_2_, and MoO_3_. Generally, Cr-containing compounds are more stable than Fe-containing compounds, and oxides and hydroxides of Fe are more likely to be dissolved in corrosion solutions [37,38]. Thus, with the increase in KCl concentration, the content of elemental Cr in the passive films increases significantly, while the content of elemental Fe decreases. Moreover, Cl^−^ ions can significantly promote the transformation of oxides into hydroxides in the passive films, due to their competitive adsorption with oxygen, so the relative contents of FeOOH and Cr(OH)_3_ tend to increase in high-concentration KCl solutions [34]. From the perspective of chemical composition, the compactness and protective capabilities of the passive films decrease with the increase in KCl concentration, leading to the deterioration of the corrosion resistance and pitting resistance of Fe-based AMCs. In previous studies, it was reported that Cr_2_O_3_, CrO_3_, and FeCr_2_O_4_ were found in p-type semiconductors, while Fe_2_O_3_, FeO, and Fe_3_O_4_ were found in n-type semiconductors [39]. Since the passive films contained oxides and hydroxides with both p-type and n-type semiconductor properties, the passive films formed on the Fe-based AMCs were bipolar.

The potentiostatic polarization curves of Fe-based AMCs at 0.3 V_SCE_ in KCl solutions with various concentrations are shown in Figure 7. In the initial stage of potentiostatic polarization, the polarization current densities of Fe-based AMCs exhibit a remarkable decline owing to the formation of passive films on the Fe-based AMCs. As the passive films become more stable, the decreasing trends of polarization current densities gradually ease or remain unchanged. In 0.05 wt% KCl solution, the polarization current densities of the Fe-based AMCs continue to decrease, indicating the improvement in the compactness of the passive films. When the KCl concentration is 1 wt%, 3 wt%, or 5 wt%, the polarization current densities of Fe-based AMCs remain stable, with little fluctuation. However, in 10 wt% KCl solution, the polarization current density fluctuates violently at a certain value and significantly increases after polarization for 21,200 s. As the KCl concentration increases to 20 wt%, the fluctuation in the polarization current density increases, and the polarization current density increases as the polarization time exceeds 13,500 s. Moreover, it should be noted that the polarization current densities of Fe-based AMCs increase with the increase in the KCl concentration. In a corrosive environment, the passive films formed on Fe-based AMCs play a crucial role in suppressing corrosion. Under low concentrations of Cl^−^ ions, compact passive films can be gradually formed on Fe-based AMCs, and the polarization current densities of Fe-based AMCs decrease with the increase in the compactness of the passive film. However, when the concentration of Cl^−^ ions is high, the compactness of the passive films decreases and metastable pitting occurs on the Fe-based AMCs, resulting in significant fluctuations in the polarization current densities. Over time, stable pitting corrosion takes place on the Fe-based AMCs, leading to a continuous increase in the polarization current densities. Therefore, similar to stainless steels or aluminum alloys, the presence of Cl^−^ ions has a destructive effect on the passive films formed on Fe-based AMCs, and an increase in the concentration of Cl^−^ ions will exacerbate the corrosion rates of Fe-based AMCs and diminish their pitting resistance [34,40].

Figure 8 illustrates the corrosion morphology of the Fe-based AMCs polarized in 3 wt% and 20 wt% KCl solutions for 10 h. In the 3 wt% KCl solution, only a few pores and inter-splat regions can be observed on the coating surfaces, indicating that Fe-based AMCs exhibit excellent corrosion resistance in low-concentration KCl solutions. As the KCl concentration increases to 20 wt%, there are some corroded splats on the coating surfaces, in addition to inherent defects. Moreover, many particles of less than 2 μm in size can be observed in these corroded splats. According to EDS analysis, these particles, enriched in elemental O, Cr, and Fe, can be identified as corrosion products, suggesting the occurrence of pitting corrosion. It is noteworthy that the corroded splats have clear boundaries; in other words, pitting corrosion only occurs on some specific splats but does not extend to the surrounding splats. Due to the inherent defects of HVOF spraying, it is difficult to avoid crystallized splats in Fe-based AMCs. These crystallized splats are susceptible to corrosion because of the potential differences between various phases and the uneven passive films [7,41]. In long-term corrosion, the crystallized splats are gradually corroded, leading to the formation of holes in the Fe-based AMCs and the infiltration of corrosion solutions.

### 3.3. Erosion Wear Behavior of the Fe-Based AMCs

To study the influence of the impact angle on the erosion rates of Fe-based AMCs, erosion wear tests were conducted on the bare and coated dissolvable Mg-RE alloy. The erosion wear profiles of the Fe-based AMCs at different impact angles are shown in Figure 9a–g. After erosion for 6 h, erosion pits formed on the Fe-based AMCs, regardless of the impact angle. With the increase in the impact angle, the worn area of the Fe-based AMCs decreased, but the maximum depth of the erosion pits initially increased and then decreased. At an impact angle of 60°, the depth of the erosion pits was the highest (5054 μm), and the dissolvable Mg-RE alloy substrate was severely damaged by erosion wear. Figure 9i presents the volumetric loss of Fe-based AMCs at different impact angles. The volumetric loss of the Fe-based AMCs was only 0.69 mm^3^ when the impact angle was 10°. The volumetric loss increased with increasing impact angle until it reached its maximum value (49.77 mm^3^) at an impact angle of 60°. With the further increase in the impact angle, the volumetric loss of the Fe-based AMCs decreased. The volumetric loss of the Fe-based AMCs was about 13.84 mm^3^ when the impact angle was 90°.

The erosion wear profiles of Fe-based AMCs and dissolvable Mg-RE alloy substrates at an impact angle of 30° are shown in Figure 9c,h, respectively. It can be seen that the worn area of the dissolvable Mg-RE alloy is far larger than that of the Fe-based AMCs, and the maximum depth of erosion pits on the substrate (1752 μm) is higher than that on Fe-based amorphous coatings (898 μm). Figure 9j shows the volumetric loss of the Fe-based AMCs and dissolvable Mg-RE alloy substrates at an impact angle of 30°: 13.68 mm^3^ and 56.96 mm^3^, respectively. Therefore, the erosion wear resistance of Fe-based AMCs is about 4.16 times higher than that of dissolvable Mg-RE alloy substrates, due to the higher hardness of the Fe-based AMCs.

Figure 10a–h illustrate the surface morphologies of the Fe-based AMCs eroded at different impact angles. The surface of uneroded Fe-based AMCs is very smooth, with only some inter-splat regions and inherent pores visible. When the impact angle is 10°, the surface of the Fe-based AMC becomes rough, and many small grooves with the same direction and some fresh splats exposed by the exfoliation of amorphous splats can be observed. As the impact angle increases to 20°, in addition to those small grooves, large grooves and plough lips emerge on the coating surfaces. It is worth noting that some broken quartz sand particles are embedded in the coating surfaces, indicating the high hardness of Fe-based AMCs. With the further increase in the impact angle, the number of large grooves continues to increase, while the number of small grooves gradually decreases. At an impact angle of 75°, in addition to large grooves and plough lips, craters and their corresponding lips appear on the coating surfaces. Notably, some broken quartz sand particles can be clearly observed in the craters. When the impact angle reaches 90°, the number of craters significantly increases, accompanied by more conspicuous deformation of amorphous splats.

The surface morphologies of Fe-based AMCs and dissolvable Mg-RE alloy substrates after erosion at an impact angle of 30° are depicted in Figure 10d,i, respectively. There are some small grooves, large grooves, plough lips, and broken quartz sand particles on the wear surface of the Fe-based AMCs. Under the same erosion conditions, many grooves, a few craters, and broken quartz sand particles can be observed on the wear surface of the dissolvable Mg-RE alloy substrates. Due to the multiphase structure of dissolvable Mg-RE alloy substrates, grooves formed by cutting appear not only on the soft α-Mg phase, but also on the hard LPSO phase. Because of their lower strength and hardness, the sizes of the grooves on the substrate surfaces is significantly higher than that of those on the coating surfaces, and craters only appear on the surface of the dissolvable Mg-RE alloy substrate.

Figure 11a–h exhibit the cross-sectional morphology of Fe-based AMCs eroded at different impact angles. Prior to erosion, the coating surface is very smooth, and some pores and inter-splat regions can be observed in the Fe-based AMCs. At an impact angle of 10°, the coating surfaces become uneven, and the surfaces of the splats are covered with dents formed by erosion. Moreover, due to the low strength of the inter-splat regions, some cracks appear among the outermost splats. These cracks initiate and subsequently propagate within the inter-splat regions, leading to the delamination of the outermost splats and the exposure of the inner splats. With the increase in the impact angle, the dents on the outermost splats become more apparent, and the number and length of the cracks along the inter-splat regions increase. At an impact angle of 45° or 60°, in addition to the cracks at the inter-splat regions, numerous fractured amorphous splats appear on the coating surfaces. As the impact angle reaches 75° or 90°, the coating surfaces display numerous craters and deformed splats, and some cracks still exist among these outermost splats. The cross-sectional morphologies of Fe-based AMCs and dissolvable Mg-RE alloy substrates after erosion at an impact angle of 30° are shown in Figure 11d,i, respectively. There are many fractured splats on the coating surfaces, and some dents caused by erosion can be observed on the outermost splats. In contrast, there are many dents caused by erosion on both the α-Mg phase and the LPSO phase, and the surface roughness of the dissolvable Mg-RE alloy substrates is much higher than that of the outmost splats.

Figure 12 illustrates the schematic diagram of the erosion wear mechanism of the Fe-based AMCs at different impact angles. When the high-speed quartz sand particles impact the Fe-based AMCs, the momentum of the quartz sand particles can be split into vertical and horizontal components, resulting in the normal stress and shear stress on the outmost splats, respectively [27]. The erosion wear mechanism of Fe-based AMCs is determined by the combined action of normal and shear stresses. At low impact angles (0–45°), the normal stress on the coating surface is relatively low, but the shear stress is comparatively high. Therefore, the Fe-based AMCs are mainly cut by quartz sand particles, leaving a quantity of small grooves on the coating surfaces. Moreover, some cracks initiate and propagate at the weak inter-splat regions under the action of repeated normal stress. When these cracks are interconnected or penetrate the inter-splat regions, the outmost splats of the Fe-based AMCs will fall off, resulting in the exposure of fresh splats. As the impact angle increases, the normal stress on the Fe-based AMCs increases, while the shear stress decreases. With the help of increased normal stress, the cutting of quartz sand particles on Fe based AMCs is intensified, and the number of large grooves on the coating surfaces increases. Moreover, due to the increase in normal stress, cracks are more likely to initiate and propagate in the inter-splat regions, leading to the aggravated delamination of the outmost splats. As the impact angle ranges from 45° to 60°, the vertical motion component of the quartz sand particles is similar to the horizontal motion component, and the outmost splats are broken under the synergistic action of normal and shear stresses. In addition, these fractured splats fall off rapidly with the help of the cracks at the inter-splat regions, resulting in the extremely high erosion rate of Fe-based AMCs. At high impact angles (60–90°), the normal stress on the coating surface is high, while the shear stress is low. Thus, the high normal velocity of the quartz sand particles leads to the severe deformation of the outmost splats and the formation of craters on the coating surface. Moreover, the lips caused by deformed splats are broken and fall off with the help of the cracks at the inter-splat regions. In summary, with the increase in the impact angle, the normal stress on the Fe-based AMCs increases, but the shear stress gradually decreases. At low impact angles (<45°), the erosion wear mechanism of Fe-based AMCs is mainly cutting and delamination. As the impact angle ranges from 45° to 60°, the erosion wear mechanism primarily involves delamination and splat fracture. At high impact angles (>60°), the erosion wear mechanism of Fe-based AMCs mainly includes delamination and deformation wear [24,27,42].

## 4. Conclusions

To address the challenge of corrosion and erosion wear in dissolvable magnesium alloy ball seats, Fe-based AMCs were prepared on dissolvable Mg-RE alloy substrates. The microstructure, corrosion behavior, and erosion wear behavior of the Fe-based AMCs were investigated, and the main conclusions are as follows:Fe-based AMCs on dissolvable Mg-RE alloy substrates can have a thickness of 1000 μm without any cracks, and the porosity and amorphous contents of Fe-based AMCs are 0.79% and 86.8%, respectively.The corrosion resistance and pitting resistance of Fe-based AMCs decrease with the increase in KCl concentration, but Fe-based AMCs exhibit a low corrosion current density of only 3.31 μA/cm^2^ and a high pitting potential of 1 V_SCE_ in 20 wt% KCl solution.Cl^−^ ions can promote the transformation of oxides into hydroxides in the passive films on Fe-based AMCs, resulting in decreases in the compactness and protective efficacy of the passive films.The erosion rates of Fe-based AMCs have a nonlinear relationship with the impact angle, initially increasing before decreasing. Specifically, the Fe-based AMCs experience the highest erosion rate when the impact angle is 60°.At an impact angle of 30°, the erosion wear resistance of Fe-based AMCs is about 4.16 times higher than that of dissolvable Mg-RE alloy substrates. The erosion wear mechanisms of Fe-based AMCs vary with the impact angles, including cutting, delamination, splat fracture, and deformation wear.Fe-based AMCs can provide excellent protection against corrosion and erosion for dissolvable Mg-RE alloy substrates. This work proposes a potential solution to protect dissolvable magnesium alloy plugging tools by thermal spraying of Fe-based AMCs.

## Figures and Tables

**Figure 1 materials-16-05170-f001:**
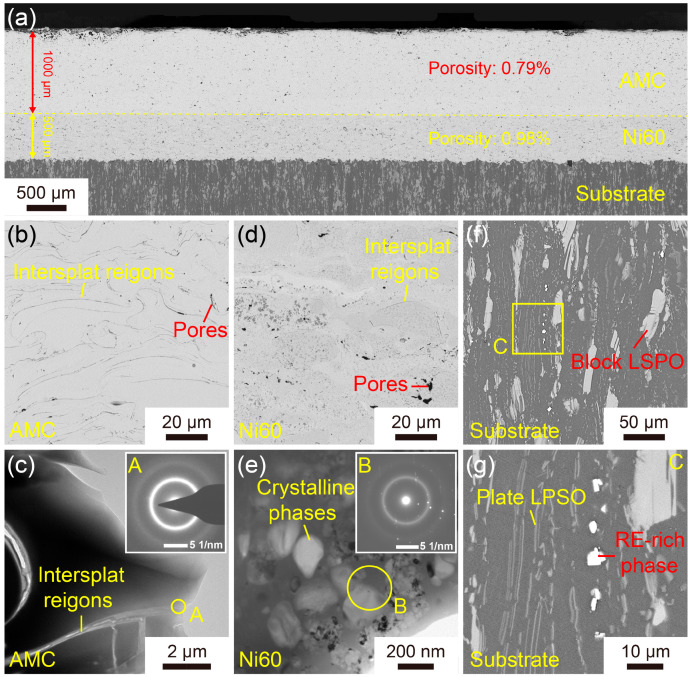
Microstructure of the Fe-based AMC on a dissolvable Mg-RE alloy substrate: (**a**–**c**) SEM and TEM images of the Fe-based AMC; (**d**,**e**) SEM and TEM images of the Ni60 interlayer; (**f**,**g**) SEM images of the dissolvable Mg-RE alloy substrate. The insets in (**c**,**e**) are the SAED patterns of the corresponding areas. A, B and C are the high-magnification images of the corresponding regions in (**c**,**e**,**f**), respectively.

**Figure 2 materials-16-05170-f002:**
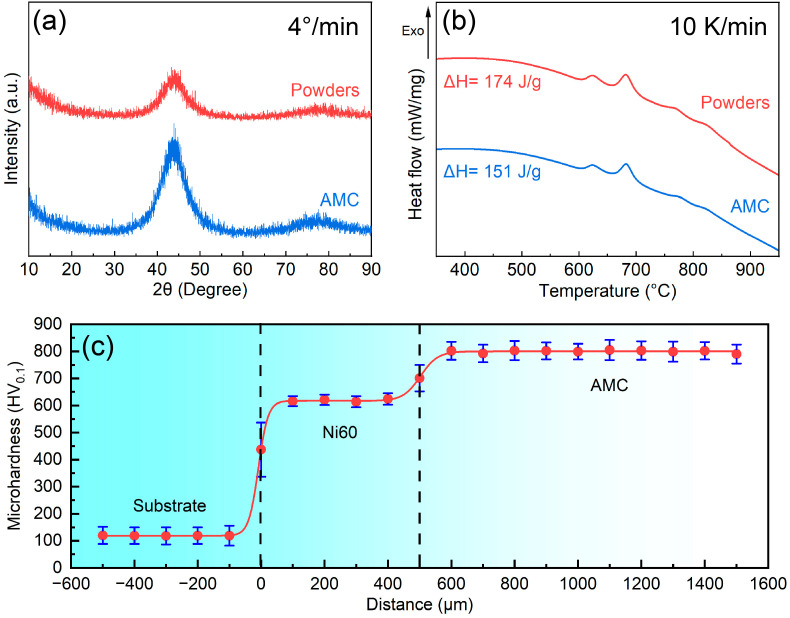
(**a**) XRD patterns and (**b**) DSC curves of the Fe-based AMC and Fe-based amorphous powders. (**c**) Microhardness of the Fe-based AMC, Ni60 interlayer, and dissolvable Mg-RE alloy substrate.

**Figure 3 materials-16-05170-f003:**
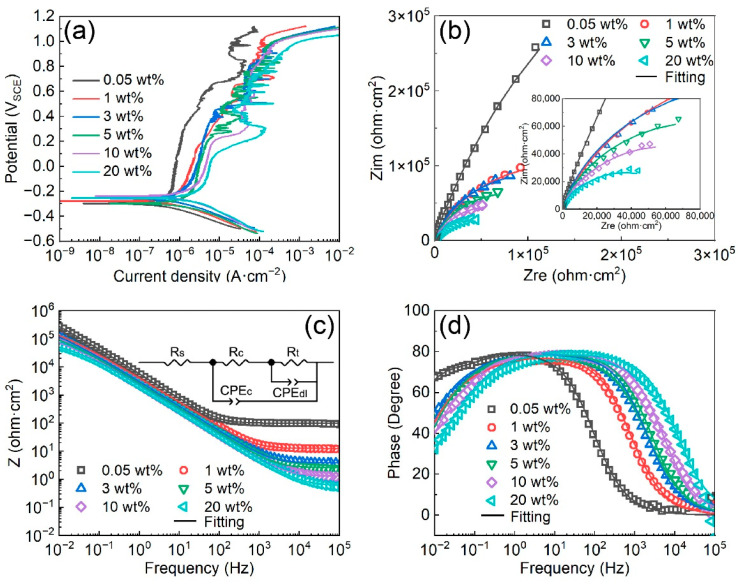
(**a**) Potentiodynamic polarization curves of the Fe-based AMCs in KCl solutions with various concentrations. (**b**) Nyquist plots, (**c**) Bode impedance plots, and (**d**) Bode phase-angle plots of the Fe-based AMCs in KCl solutions with various concentrations. The inset in (**c**) is the equivalent circuit of Fe-based AMCs.

**Figure 4 materials-16-05170-f004:**
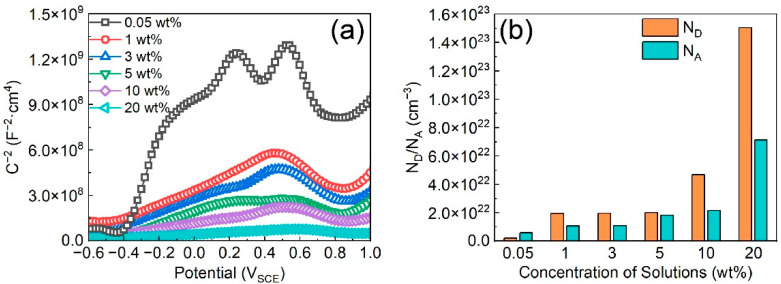
(**a**) Mott–Schottky plots and (**b**) carrier concentrations of the passive films formed on Fe-based AMCs in KCl solutions with various concentrations.

**Figure 5 materials-16-05170-f005:**
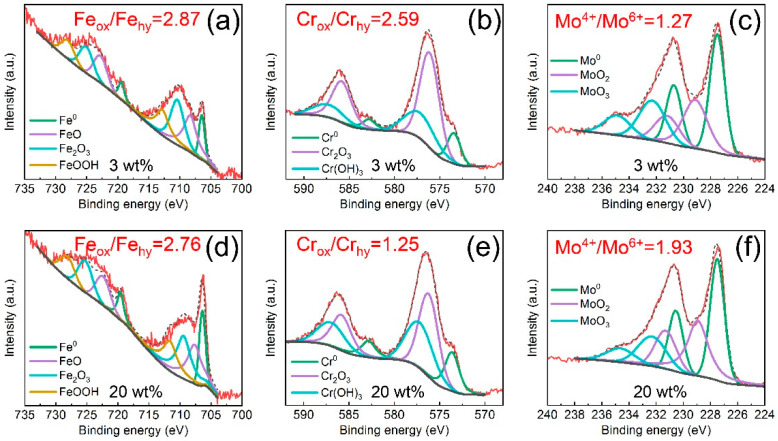
XPS spectra for the Fe-based AMCs in KCl solutions with various concentrations: (**a**–**c**) 3 wt%, (**d**–**f**) 20 wt%.

**Figure 6 materials-16-05170-f006:**
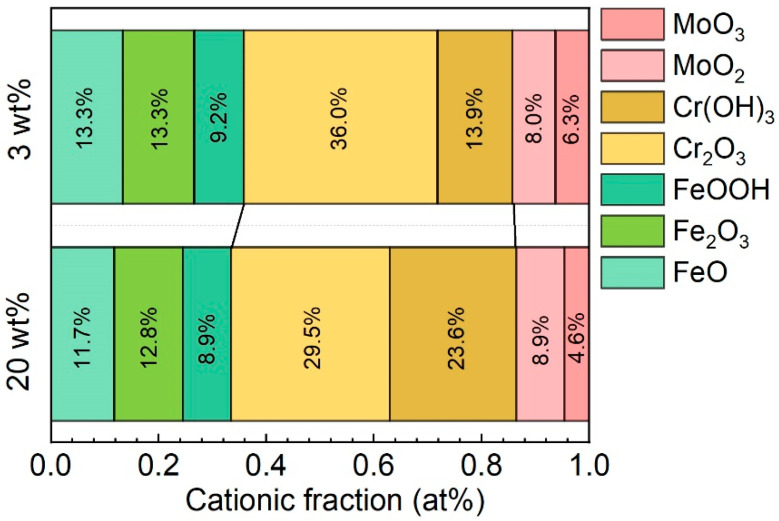
Chemical compositions of the passive films formed on Fe-based AMCs in KCl solutions with various concentrations.

**Figure 7 materials-16-05170-f007:**
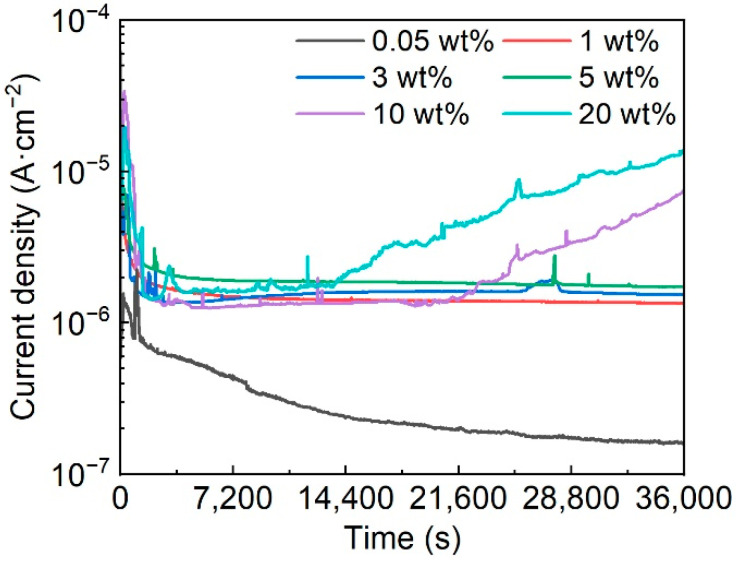
Potentiostatic polarization curves of the Fe-based AMCs at the potential of 0.3 V_SCE_ in KCl solutions with various concentrations.

**Figure 8 materials-16-05170-f008:**
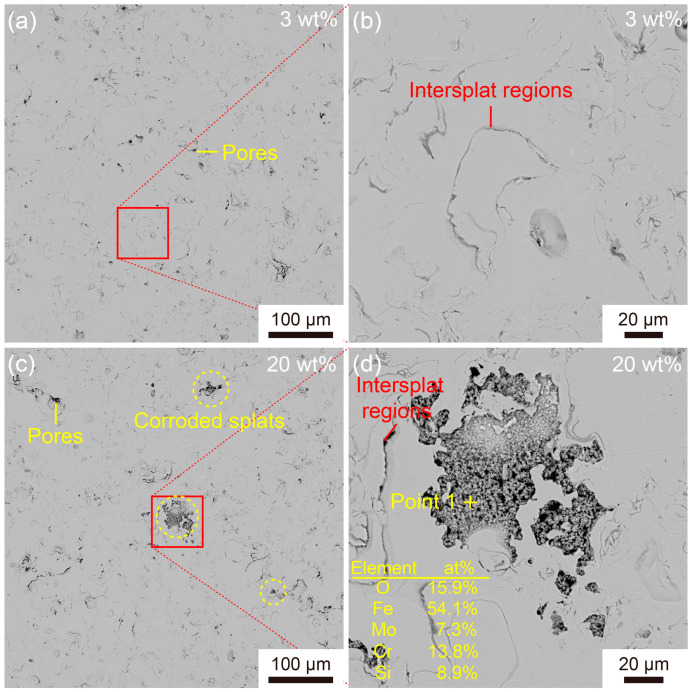
Surface SEM images of Fe-based AMCs after polarization for 10 h in KCl solution with various concentrations: (**a**,**b**) 3 wt%, (**c**,**d**) 20 wt%.

**Figure 9 materials-16-05170-f009:**
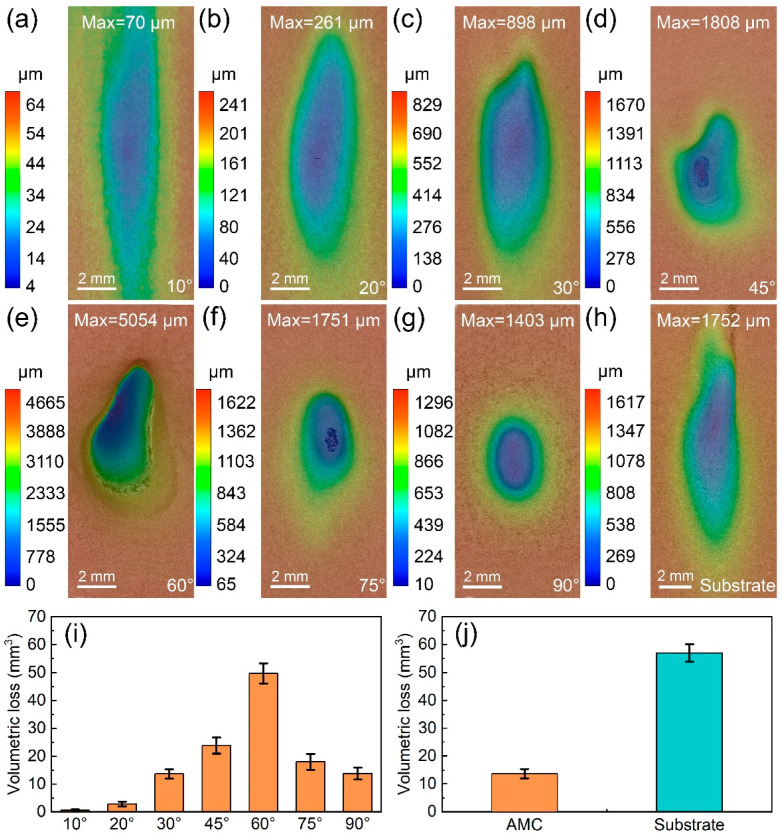
Erosion wear profiles of the Fe-based AMCs at different impact angles: (**a**) 10°, (**b**) 20°, (**c**) 30°, (**d**) 45°, (**e**) 60°, (**f**) 75°, and (**g**) 90°. (**h**) Erosion wear profiles of the dissolvable Mg-RE alloy substrates at an impact angle of 30°. (**i**) Volumetric loss of Fe-based AMCs at different impact angles. (**j**) Volumetric loss of Fe-based AMCs and dissolvable Mg-RE alloy substrates at an impact angle of 30°.

**Figure 10 materials-16-05170-f010:**
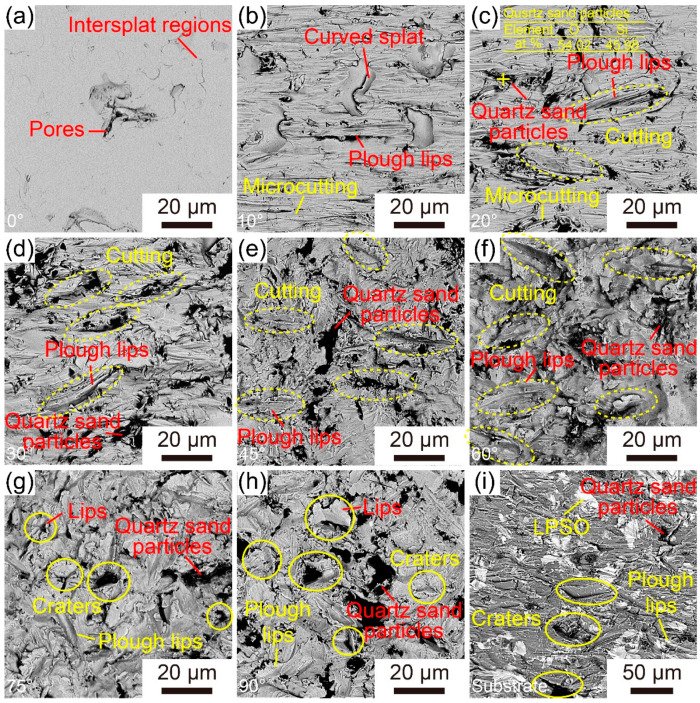
Surface SEM images of the Fe-based AMCs eroded at different impact angles: (**a**) 0°, (**b**) 10°, (**c**) 20°, (**d**) 30°, (**e**) 45°, (**f**) 60°, (**g**) 75° and (**h**) 90°. (**i**) Surface SEM images of the dissolvable Mg-RE alloy substrates eroded at an impact angle of 30°.

**Figure 11 materials-16-05170-f011:**
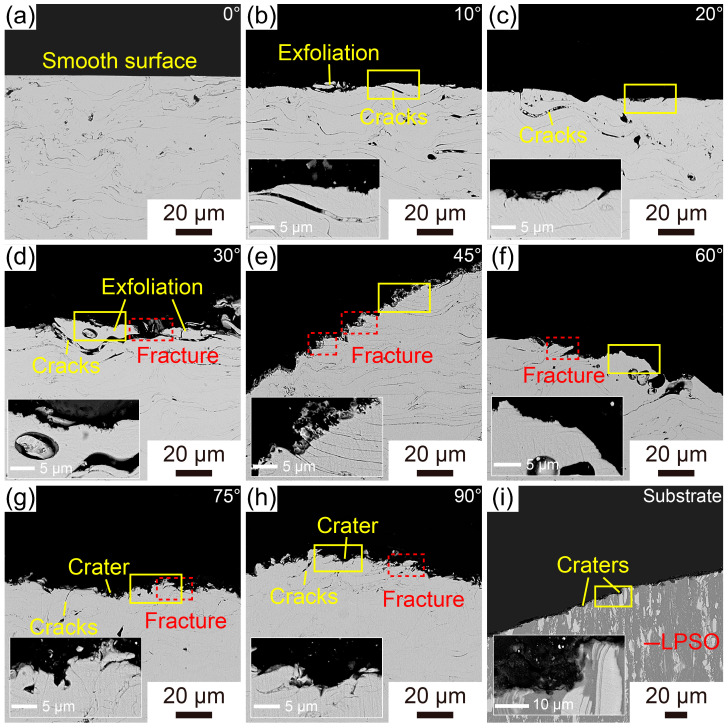
Cross-sectional SEM images of the Fe-based AMCs eroded at different impact angles: (**a**) 0°, (**b**) 10°, (**c**) 20°, (**d**) 30°, (**e**) 45°, (**f**) 60°, (**g**) 75° and (**h**) 90°. (**i**) Cross-sectional SEM images of the dissolvable Mg-RE alloy substrates eroded at an impact angle of 30°.

**Figure 12 materials-16-05170-f012:**
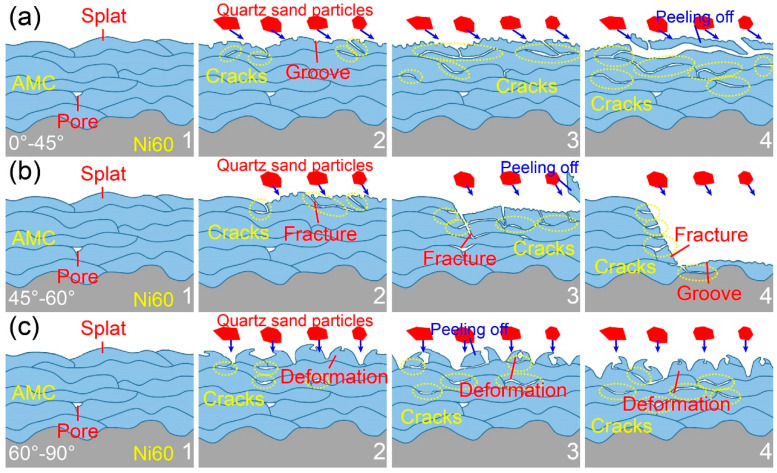
Schematic diagram of the erosion wear mechanism of Fe-based AMCs at different impact angles: (**a**) 0–45°, (**b**) 45–60°, and (**c**) 60–90°.

**Table 1 materials-16-05170-t001:** HVOF spraying parameters for Fe-based AMCs and Ni60 interlayers.

Coatings	Fe-Based AMCs	Ni60 Interlayers
Kerosene flow rate (L/s)	0.0074	0.0063
Oxygen flow rate (m^3^/s)	0.0149	0.0146
Spray distance (mm)	320	330
Powder feed rate (g/s))	0.6667	1.3333
Scanning velocity (mm/s)	300	300

**Table 2 materials-16-05170-t002:** Corrosion parameters of Fe-based AMCs obtained from the potentiodynamic polarization curves.

Samples	E_corr_ (V_SCE_)	I_corr_ (μA/cm^2^)	E_pit_ (V_SCE_)	I_pit_ (μA/cm^2^)
0.05 wt%	−0.296	0.64	1.002	29.71
1 wt%	−0.288	1.16	1.008	162.56
3 wt%	−0.287	1.66	1.003	189.23
5 wt%	−0.292	2.06	1.002	193.51
10 wt%	−0.257	2.68	1.007	266.45
20 wt%	−0.274	3.31	1.002	648.78

**Table 3 materials-16-05170-t003:** Fitting parameters of Fe-based AMCs obtained from the EIS data.

Samples	0.05 wt%	1 wt%	3 wt%	5 wt%	10 wt%	20 wt%
R_s_ (Ω·cm^2^)	99.54	12.43	3.96	2.72	1.43	0.77
R_c_ (Ω·cm^2^)	337,440	107,500	98,060	93,491	59,629	38,112
CPE_c_ (S·s^n^·cm^−2^)	3.38 × 10^−5^	5.03 × 10^−5^	6.53 × 10^−5^	7.61 × 10^−5^	8.58 × 10^−5^	9.09 × 10^−5^
CPE_c_-n	0.891	0.865	0.879	0.879	0.882	0.877
R_t_ (Ω·cm^2^)	999,910	144,420	112,630	66,516	56,394	31,655
CPE_dl_ (S·s^n^·cm^−2^)	1.06 × 10^−5^	3.24 × 10^−5^	3.89 × 10^−5^	7.51 × 10^−5^	9.68 × 10^−5^	1.42 × 10^−4^
CPE_dl_-n	0.869	0.841	0.834	0.824	0.846	0.845

## Data Availability

Data are contained within the article and can be requested from the corresponding author.

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
