# Peer review of "Corrosion and Erosion Wear Behaviors of HVOF-Sprayed Fe-Based Amorphous Coatings on Dissolvable Mg-RE Alloy Substrates"

_materials, 2023, doi:10.3390/ma16145170_

Round 1
Reviewer 1 Report
The present manuscript by Jun Yang, Yijao Sun, Minwen Su, Xueming Yin, Hongxiang Li, and Jishan Zhang, entitled "Corrosion and wear behaviors of HVOF-sprayed Fe-based amorphous coating on dissolvable Mg-Re alloy substrates" highlights the study on resistance to corrosion, as well as to erosion wear of Fe-based (Fe48.8Cr23.4Mo19.8Si5C2.1B0.9) amorphous high velocity oxy-fuel sprayed (further HVOFS) coatings. Despite thermally sprayed Fe-based coatings have been studied for decades, by the reviewer's best knowledge, the present results haven't been published yet. Thus this research may be considered novel. By the reviewer's estimation, the object of the study is highly relevant, and the results of the study are of a high interest as for scientists, as for engineers. Moreover, the volume of the completed research is quite impressive, too.
The manuscript itself leaves a nearly perfect impression. The manuscript title entirely corresponds with its contents. The abstract summarizes all the principle findings in a consice, but comprehensive manner. The chapter 'Introduction' gives a very detailed overview of the object of the research and the research background, presenting the idea of the study in a logical and convincing way. All the experimental methods, described in the chapter 'Materials and Methods', are entirely relevant and do allow to perform a holistic analysis, although the description of the experimental activities could contain more details (the reviewer's exact recommendations are given below). The results are presented clearly and logically. The analysis is very thorough and explicit. The conclusions are short, but do reflect all the main points. The number of references is quite significant. It is worth noting that approximately 43% of them (eighteen references out of 42) are from 2020 or later years, what is an indirect indication of a high relevance of this research. English is nearly errorless, except for few typos (specified below), and so is the formatting.
To conclude, the reviewer recommends to accept this paper for publication. At the same time, to make the manuscript completely perfect, he would recommend to introduce few minor corrections, which are listed below, during the proofreading stage.
COMMENTS TOWARDS THE TEXT.
1. Materials and Methods -> Table 1: in general, L/h, m3/h and g/min could be recalculated to L/s, m3/s and g/s (as hours and minutes aren't SI units).
2. Lines 114-116: scan range and step of measurements, applied in the XRD procedure, could be specified.
3. Line 134: '(0.98 N)' should be inserted after '100 g' (as g (otherwise designated gf) isn't a SI unit).
4. Lines 147-154: the impact angles, abrasive type, abrasive average size, abrasive amount and velocity must be specified here.
5. Line 174: the abbreviation 'LPSO' should be explained.
6. Line 374: was the deepest pit really 5045 μm (=5.045 mm) deep? In such a case, the wear was very intensive, and the pit would penetrate deep into the substrate.
7. Line 385: was the deepest erosion pit, which formed under the impact angle of 30° in the Mg-RE substrate really 1752 μm (=1.752 mm) deep?
8. Line 386: was the deepest erosion pit, which formed under the impact angle of 30° in the Fe-based amorphous coating really 898 μm (=0.898 mm) deep?
LANGUAGE AND FORMATTING REMARKS.
1. Line 24: 'reaches maximum' or 'reaches the highest value', not simply 'reaches the highest'.
2. Line 57: probably 'low thickness', not 'thin thickness'.
3. Lines 124-125: probably 'interlayers were reached', not 'interlayers were achieved'.
4. Tilda (~) should be replaced by hyphen (-) in
a) Line 185;
b) Line 187;
c) Line 454;
d) Line 470;
e) Line 483.
5. 'Rc and Rt' should replace 'Rc and Rt' in
a) Line 234;
b) Line 235;
c) Line 237;
d) Line 238.
6. Line 240: 'CPEc and CPEdl', not 'CPEc and CPEdl'.
7. Line 293: 'Feox/Fehy', not 'Feox/Fehy'.
8. Line 295: 'Crox/Crhy', not 'Crox/Crhy'.
9. 'quarz sand particles' or 'quarz sand grains' should apparently be used instead of 'quarz sands' in
a) Line 404;
b) Line 408;
c) Line 414;
d) Line 416;
e) Line 450;
f) Line 451;
g) Line 456;
h) Line 462;
i) Line 462;
j) Line 466;
k) Line 472;
l) Figure 12.
The reviewer finds that the presented manuscript may be accepted for publicaiton.
Reviewer 2 Report
Overall, the quality of the paper is good. However, there are some minor corrections to be made, specially in the item 3.3. (Erosion wear behaviour of the Fe-based AMC).
1. Clearly the outer surface of the coating were grinded before erosion test, but that fact was not mentioned in the Methodology section.
2. In the Figure 10, there is no evidence that quartz sand are embedded in the substrate. An EDS analysis are required to prove that.
3. Use the word "groove" instead of "furrows".
4. In the Figure 12, please indicate the direction/angle of impact of the quartz sand particles shown.
5. The complete chemical composition of the bond coat layer is missing.
The paper needs to be revised by an native english writer.
